# Global Compressive Loading from an Ultra-Thin PEEK Button Augment Enhances Fibrocartilage Regeneration of Rotator Cuff Enthesis

**DOI:** 10.3390/bioengineering10050565

**Published:** 2023-05-09

**Authors:** Chia-Wei Lin, En-Rung Chiang, Shih-Hao Chen, Poyu Chen, Heng-Jui Liu, Joe Chih-Hao Chiu

**Affiliations:** 1Orthopedic Department, Wuri Lin Shin Hospital, Taichung 414, Taiwan; cutemo0953@gmail.com; 2Department of Clinical Research, De Novo Orthopedics Inc., Taichung 414, Taiwan; 3Department of Orthopaedics and Traumatology, Taipei Veterans General Hospital, Taipei 112, Taiwan; rondoo0614@gmail.com; 4Department of Surgery, School of Medicine, National Yang-Ming University, Taipei 112, Taiwan; 5Department of Orthopedic Surgery, Taichung Tzu Chi Hospital, Buddhist Tzu Chi Medical Foundation, Taichung 427, Taiwan; shihhao603@tzuchi.com.tw; 6Department of Orthopaedics, Tzu-Chi University, Hualien 970, Taiwan; 7Department of Orthopedic Surgery, Linkou Chang Gung Memorial Hospital, Taoyuan 333, Taiwan; poyui.chen@gmail.com; 8Department of Occupational Therapy, Graduate Institute of Behavioral Sciences, College of Medicine, Chang Gung University, Taoyuan 333, Taiwan; 9Healthy Aging Research Center, Chang Gung University, Taoyuan 333, Taiwan; 10Bone and Joint Research Center, Linkou Chang Gung Memorial Hospital, Taoyuan 333, Taiwan; 11Comprehensive Sports Medicine Center (CSMC), Linkou Chang Gung Memorial Hospital, Taoyuan 333, Taiwan

**Keywords:** rotator cuff repair, augmentation, footprint compression, transosseous equivalent, double row

## Abstract

A PEEK button is developed to improve the tendon-to-bone compression area. In total, 18 goats were divided into 12-week, 4-week, and 0-week groups. All underwent bilateral detachment of the infraspinatus tendon. In the 12-week group, 6 were fixed with a 0.8–1 mm-thick PEEK augment (A-12, Augmented), and 6 were fixed with the double-row technique (DR-12). Overall, 6 infraspinatus were fixed with PEEK augment (A-4) and without PEEK augment (DR-4) in the 4-week group. The same condition was performed in the 0-week groups (A-0 and DR-0). Mechanical testing, immunohistochemistry assessment, cell responses, tissue alternation, surgical impact, remodeling, and the expression of type I, II, and III collagen of the native tendon-to-bone insertion and new footprint areas were evaluated. The average maximum load in the A-12 group (393.75 (84.40) N) was significantly larger than in the TOE-12 group (229.17 (43.94) N) (*p* < 0.001). Cell responses and tissue alternations in the 4-week group were slight. The new footprint area of the A-4 group had better fibrocartilage maturation and more type III collagen expression than in DR-4 group. This result proved the novel device is safe and provides superior load-displacement to the double-row technique. There is a trend toward better fibrocartilage maturation and more collagen III secretions in the PEEK augmentation group.

## 1. Introduction

The success of rotator cuff repair depends on both intrinsic musculotendinous qualities and the surgical technique [1,2,3,4]. Plenty of surgical techniques have been developed to provide minimal gap formation, a high initial fixation strength, and the maintenance of mechanical stability until tendon-to-bone healing, characterized by the presence of fibrocartilage tissue connecting cuff structures to deeper layers of the bone [5,6,7,8,9,10,11,12,13,14,15]. However, scar tissue, instead of the native enthesis, is commonly deposited at the healing interface and has been identified as a potential cause of the high failure rate of cuff repair because of the weak mechanical properties and less mineralized fibrocartilage within. A good compression against the rotator cuff footprint, while maximizing the biological factors that allow ultimate tendon-to-bone healing, has been suggested to provide good cuff healing [16,17,18,19,20,21]. In the present study, we developed a novel fixation method for rotator cuff repair utilizing an ultra-thin polyetherketone (PEEK) button used to improve the area and extent of tendon-to-bone compression, as PEEK material is oftentimes used in rotator cuff repairs because it is biologically inert, radiolucent, and resistant to hydrolysis and oxidation [22,23,24,25,26,27]. We hypothesized that this method is safe and would result in a superior enthesis regeneration, characterized by greater fibrocartilage formation and improved collagen fiber organization in an acute rotator cuff tear goat model. This PEEK button augmentation would lead to higher biomechanical stiffness when compared with a simple repair with sutures.

## 2. Materials and Methods

### 2.1. Study Design and Surgery Techniques

All animal work was conducted following a project license protocol accepted under the Institutional Animal Care and Use Committee (IACUC). All authors adhered to and included the ARRIVE checklist. A total of 18 male black goats of 40 kg (approximately 52 weeks of age) were divided into 3 groups: 12-week, 4-week, and 0-week (sacrificed right after the surgery). All goats underwent bilateral detachment of the infraspinatus tendon. In the 12-week group, there were 6 goats’ infraspinatus tendon fixed with a PEEK augment for right shoulders (A-12, augment) and the double-row (DR) suture technique from anchors for left shoulders (DR-12) without a PEEK augment. In the 4-week group, infraspinatus tendons were fixed with suture anchors, and 1 PEEK augment for every right shoulder of 6 goats (A-4). Another six left shoulders were fixed with two suture anchors as DR configuration (DR-4). In the 0-week group, infraspinatus tendons were fixed with suture anchors, and 1 PEEK augment for every right shoulder of 6 goats (A-0). Another six left shoulders were fixed with two suture anchors (DR-0).

In the A-12 groups, one double-loaded suture anchor was inserted in the medial row first. The sutures were then passed through the torn tendon and the holes of the PEEK augment. After knot tying, four stitches were pulled laterally and fixed into greater tuberosity, with one knotless anchor as the lateral row fixation. The same procedure was carried out in the A-4 and A-0 groups. In the DR-12 group, the same fashion of anchor placement was performed without the PEEK augment, mimicking the commonly used DR technique [28]. In the DR-4 and DR-0 groups, the infraspinatus tendons were repaired with two double-loaded suture anchors in a double-row manner and knotted at the medial and lateral edges of the tendon footprint. All suture configurations are demonstrated in Figure 1. The A-12, A-4, and A-0 groups served as experiment groups, while DR-12, DR-4, and DR-0 served as control groups.

### 2.2. Implant Design

The PEEK augment (Figure 2a) is designed to yield a global compressive load on the footprint of the cuff. It is 0.8–1 mm thick and highly elastic to fit humerus geometry well. The medial concave is designed to minimize the strangulation of the blood circulation around the musculotendinous junction of the cuff. The lateral concave of the PEEK augment avoids the impingement of greater tuberosity during shoulder abduction. The compression force on the medial side decreases when the stitches through eyelets are pulled laterally for lateral-row fixation because of the see-saw effect (Figure 2b), which decreases circulation compromise. Anterior and posterior bended fins were designed to provide compression force onto the rotator cable (Figure 2c). The PEEK augment provides more evenly distributed compression force on the rotator cuff footprint according to the fine element analysis (Figure 2d,e). We used a 5.5 mm Healix Peek anchor (DePuy Mitek, Raynham, MA, USA) for the medial row and a 4.75 mm PEEK SwiveLock suture anchor (Arthrex, Naples, FL, USA) for the lateral row fixation. Horizontal mattress sutures were tied using five alternating half-hitch knots to reproduce arthroscopic knot configurations. This represented the commonly used double-row suture-bridge technique popularized in clinical practice and provided the expected fixation strength.

### 2.3. Biomechanical Testing

The mechanical experiment was performed as Liu et al. proposed [29]. The HT-2402EC material testing machine was employed in the study. The test equipment characteristics’ definitions are listed in Table 1.

In the 12-week and 0-week groups, the humerus was fixed in the embedding fixture (Figure 3a,b). The infraspinatus muscle was erected 90 degrees from the humerus and pulled perpendicular to the sagittal plane of the fixture, as shown in Figure 3c.

The test machine provided a continuous pulling force at a 75 mm/min rate until failure of the infraspinatus muscle (Figure 4a–c). The experiment stopped until the tensile force decreased to 0. Damage to the test specimen was assessed after each test and photographed. The maximum pull force (N) and load displacement (mm) during the study were recorded. In the 0-week group, the specimens were harvested right after the surgery, and the same experimental setup was applied.

### 2.4. Immunohistochemistry (IHC) Assessment

The proximal humerus was removed to survey the fibrocartilage growth onto the footprint of the repaired infraspinatus following mechanical testing. Each sample was preserved in 10% neutral buffered formalin (NBF) at room temperature. After 48 to 72 h of NBF fixation, the samples were trimmed and processed for decalcification, followed by wax infiltration. Each sample was fixed with 10% formal saline and underwent decalcification in EDTA, ascending graded alcohol dehydration, and defatting in chloroform and was then embedded in paraffin. Multiple 4 mm-thick slides were cut in the coronal plane through the humerus, enthesis, and infraspinatus musculotendinous unit before staining with hematoxylin and eosin and Masson’s trichrome stain [30], followed by pathologist examination. Two blinded observers evaluated all sections using a BH-2 light microscope (Olympus).

### 2.5. The Safety Assessment of Cell Response after Surgical Repairs

The first area of interest (AOI) was targeted on the tendon-to-bone insertion regarding cell response with implanted materials, evaluated with the histopathological standard through an H&E stain. The scoring system is listed in Table 2.

### 2.6. The Safety Assessment of Tissue Alternation after Surgical Repairs

The 2nd AOI was focused on the adherent tissue around the implanted PEEK augment around the tendon-to-bone insertion. The histopathology evaluation format for implant material scoring through section was applied via H&E stain and Masson’s trichrome stain. The scoring system is listed in Table 3.

### 2.7. Overall Rating of Surgical Impact toward Rotator Cuff

The 3rd AOI was focused on the surgical impact toward the rotator cuff. An overall rating of test samples was given using a rating range of 0 to 15, as shown in Table 4.

### 2.8. The Remodeling of Native Tendon-to-Bone Insertion and New Footprints Area

The 4th AOI was concentrated on the native tendon-bone and new footprint area interface that underwent the surgical approach and regeneration. The maturation of the enthesis was assessed according to a semi-quantitative scoring system developed by Ide et al. [31], as shown in Table 5.

### 2.9. The Expression of Type I, II, III Collagen in New Footprint Area

The 5th and 6th AOI aimed at collagen deposition and proteoglycan contents alternation in the new/old footprint area along the fibrocartilage zone; the segments were divided into bone, cartilage, and tendon parts for observations between the A-4 and DR-4 groups.

### 2.10. Statistical Analysis

The recorded data for fibrocartilage maturation scores were calculated by the Mann–Whitney U test and the Fisher exact test through two-tailed hypothesis between the A-4 and DR-4 groups. The observed result for the IHC stain was analyzed by the Fisher exact test through identical hypotheses among three groups. The difference of load displacement was determined by one-way ANOVA. The differences of *p* < 0.05 were considered significant via software configuration (GraphPad Prism version 9.0.1 for MacOS, GraphPad Software GraphPad Software Inc., La Jolla, CA, USA).

## 3. Results

All animals survived during the study without infection. Limping was noted for the first 3 to 5 postoperative days, but a normal gait pattern returned afterward.

### 3.1. Macroscopic Findings

At the time of euthanasia, continuity between the repaired tendon and bone was observed in all A-12 and DR-12 groups. No anchor was pulled out. In A-4 and DR-4 groups, all specimens were cut at the infraspinatus muscle part at the final timepoint. The specimens in the A-4 group showed the well-developed fibrotic appearance of the enthesis, as shown in Figure 5a. An atrophic and retracted native cuff tendon and newly formed enthesis, lying laterally between the augment and knotless anchor, were observed and are shown in Figure 5b. A new enthesis of DR-4 specimens was also observed, but was thinner (3.5 mm, Figure 5c) than that of the A-4 group (7.5 mm) in gross appearance.

### 3.2. Biomechanical Testing 

The load-displacement curve of the study is shown in Figure 6. The graph shows that the force decreased after pulling to the maximum load, until infraspinatus muscle failure occurred (when the tensile force achieved 0). The pull-out strength of each group is shown in Table 6. The average maximum load in the A-12 group (393.75 (84.40) N) was significantly larger than that of the DR-12 group (229.17 (43.94) N) (*p* < 0.001). There was no significant difference in the maximum load in the A-0 group and DR-0 group (maximum load, 102.98 (23.14) N, and 94.32 (29.32) N, *p* = 0.291). All specimens in the A-12 and DR-12 groups had tears at the muscular part of the infraspinatus, while all specimens in the A-0 and DR-0 groups had a tear at the enthesis. No suture breakage or anchor pull-out was observed. This result implied that the PEEK augment significantly improved the tendon healing quality and provided superior load displacement in the A-12 group compared to the DR-12 group at a 12-week time interval, which was not observed at time zero (A-0 and DR-0 groups).

### 3.3. Histologic Findings

In the safety assessment of cell response and tissue alternation after surgical repair, the tendon-to-bone insertion of the A-4 group showed a slight inflammatory reaction compared to that of the DR-4 group. The score of the cell response and tissue alternation of the A-4 group was 99 (average, 16.5). Those of the DR-4 group were 72 (average, 12). The difference between the average cell response and tissue alternation score was 4.5, classified as a slight reaction. The details are listed in Figure 7 and Table 7. The PEEK augment seemed to induce a more sub-acute to sub-chronic inflammatory response, such as fibrin deposition and cell apoptosis. Therefore, the PEEK augment might arouse higher irritation than stitches alone. The low- and high-power fields section of both the A-4 and DR-4 groups is shown is Figure 8.

The native footprint area had similar fibrocartilage formation in both the A-4 and DR-4 groups (Enthesis maturation grade ≥ 3, A-4: 3/6, 50%; DR-4: 1/6, 16.7%; *p* = 0.545). The cartilage cells’ integrity was absent to sporadic, distributed in the native footprint part. However, the new footprint area of the A-4 group had better fibrocartilage maturation (Enthesis maturation grade ≥ 3, 5 of 6, 83.3%) than that of the DR-4 group (3 of 6, 50%). Though there was a trend that the A-4 group had a better fibrocartilage maturation score than the DR-4 group did, no statistical significance was achieved. The result of tissue remodeling is shown in Table 8.

In the IHC stain within the native footprint area, both the A-4 and DR-4 groups revealed an identical tendency among collagen I/II/III on tendon, cartilage, and bone (Table 9).

Within the new footprint area of the A-4 group, type III collagen was widely found in the tendon (4 out of 6) and cartilage (3 out of 3). Type I collagen was positive in the cartilage of only one specimen in the A-4 group (Table 10). The H&E stain and IHC stain in the new footprints area of the A-4 group and DR-4 groups are shown in Figure 9.

Since collagen III acts as a major extracellular matrix element prior to tendon regeneration and maturation [32], the IHC results in the current investigation implied that the PEEK augment accelerated the healing process. Better fibrocartilage growth with more collagen III secretion was observed in the new footprint of the A-4 group than in the DR-4 group.

In summary, the PEEK augment provided homogeneous pressure distribution on the cuff footprint, contributing to better fibrocartilage growth, with more collagen III secretions in the new footprint of the A-4 group than in the DR-4 group. At 12 weeks, the PEEK augment group (A-12) had an improved tendon healing quality and superior load displacement compared to the DR-12 group.

## 4. Discussion

In our study, the safety and effectiveness of the PEEK augment were evaluated. From the result of the A-12 and DR-12 groups, the biomechanical stiffness of the A-12 group was significantly higher than that of the DR group for 71.8%. In the A-4 group, better fibrocartilage maturation and more positive type III collagen in tendon and bone was observed than in the DR-4 group. This implies the PEEK augment provides benefits for enthesis healing via its biomechanical characteristics.

Double-row repair provides better footprint compression than single-row repair does [28]. In our study, the PEEK augment is even more beneficial than the double-row repair when comparing the early histology result between the A-4 and DR-4 groups. Slight inflammation was observed in the A-4 group, which implies the PEEK augment has a limited adverse effect on the cuff tendon surface. Though not statistically significant, the fibrocartilage maturation score at the enthesis was better in the A-4 group than in the DR-4 group. An IHC stain revealed positive type III collagen only in the cartilage and tendon in the A-4 group, which suggested the augment may provide a better environment for the healing of a rotator cuff tendon than DR repair does.

Whether in the A-4 or the DR-4 group, new enthesis was formed at the area lateral to the native footprint. Fibrocartilage maturation was also less obvious within the native cuff tendon and its tendon-to-bone interface than in the new enthesis. Most surgeons apply biologic materials at the native cuff footprint rather than the lateral part. However, Thon et al. had a similar concept to our study. They applied a bioinductive collagen patch to the most lateral part of a repaired cuff tendon in large or massive cuff tear patients, secured with PEEK bone staples. The result showed a relatively higher healing rate at the 2-year follow-up. They suggested that in addition to the collagen matrix implant possibly promoting blood flow and tissue healing, its biomechanical benefit to the area lateral to the native cuff tendon should also be considered [33]. Thon et al. then claimed in a review article that the use of the Regeneten ( Smith & Nephew, Andover, MA, USA) implant on top of the repaired cuff showed improved patient-reported outcomes and success when compared to isolated cuff repair without Regeneten augmentation [34].

However, the mechanism of why the PEEK augment yielded better healing potential remained unclear in this study. A study from Cole et al. [35] found there was a decrease in re-tear rates from a single-row repair to a double-row repair to transosseous-equivalent repairs. In the current study, we compared the effects of sutures and the PEEK augment on cuff healing and implied that the PEEK augment provided better enthesis healing, which might be because of the more even compression force provided by the PEEK augment design. The PEEK augment can transform the tensile force of the sutures into compression force, which forms a cyclic “actuator-like” compressive load, stimulating fibrocartilage transformation, as healing of the rotator cuff tendon requires biomechanical fixation that provides adequate strength, stability, and compression against the rotator cuff footprint, while maximizing the biologic factors that allow ultimate tendon-to-bone healing [16].

In biomechanical tests of the A-0 and DR-0 groups, the results showed no significant difference between the two groups. There are several studies comparing knotted or knotless medial row constructs with conflicting results. In our study, there were no marked differences in failure modes between the A-0 and DR-0 groups. On the medial aspect of the whole construct, the knots were on top of the PEEK augment rather than the bursal side cuff surface, where stress concentration was impeded. When the sutures are pulled laterally, the see-saw effect prevents medial side strangulation and failure, which is evident in the suture bridge fixation technique that leads to a 45% reduction in blood flow when compared with only medial row fixation [36].

There are limitations regarding this study. First, the delivery of the patch and biologic material with sutures under arthroscopic rotator cuff repair remains a technical challenge and requires longer surgical time. Further study should focus on how to apply the PEEK augment along with rotator cuff repair arthroscopically in a relatively easy and efficient way. The second limitation of this study is the small case number. Only six shoulders were included in each group, which might result in the insignificance of the maturation score between the A-4 and DR-4 groups, despite the trend. Further study with a larger number in each group and more groups with a longer period of follow-up might also be helpful to elucidate the role of the PEEK augment in facilitating rotator cuff healing.

## 5. Conclusions

In the present study, we proved the safety and effectiveness of the ultra-thin PEEK button augment used during open rotator cuff repair in a goat model. This provides a superior load-displacement property compared to the conventional DR technique. Though not significantly different, there is a trend toward better fibrocartilage maturation and more collagen III secretion in the PEEK augmentation group than in the conventional DR repair group. The PEEK augment could be further applied to increase footprint compression and avoid medial-row blood supply strangulation during rotator cuff repair.

## Figures and Tables

**Figure 1 bioengineering-10-00565-f001:**
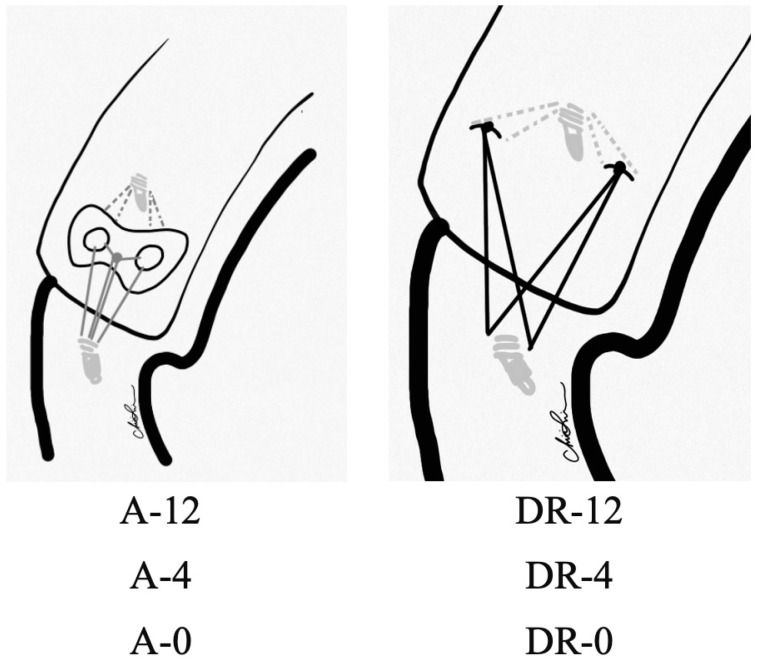
Experimental groups. A: Augment; DR: Double-row.

**Figure 2 bioengineering-10-00565-f002:**
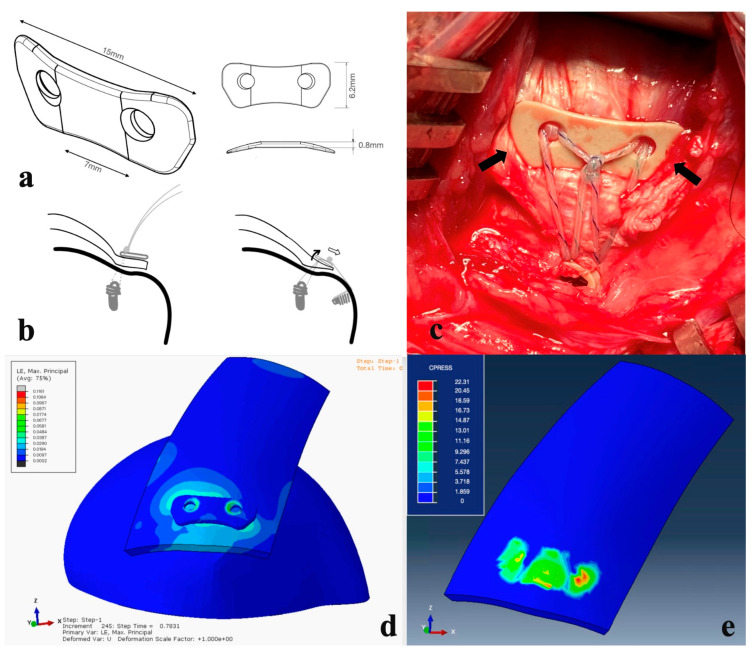
(**a**) Perspective view of the PEEK augment. (**b**) See-saw effect. When the sutures are pulled laterally (right), more compressive force is applied on the lateral side and loosened at the medial part of the PEEK augment, preventing circulation compromise. (**c**) In the A-4 group, the PEEK augment was applied onto the rotator cuff. Bilateral fins (arrows) yielded compressive force onto the rotator cable and fit local anatomic structures well. (**d**,**e**) The pressure is dispersed averagely underneath the PEEK button, while the sutures used during rotator cuff repairs are thin and the stress concentration lies in the suture-tendon area. The PEEK material sustains the shear force of the sutures, converting them into a compression force and providing a homogeneous pressure distribution, which could further benefit the tendon healing proved in our animal study.

**Figure 3 bioengineering-10-00565-f003:**
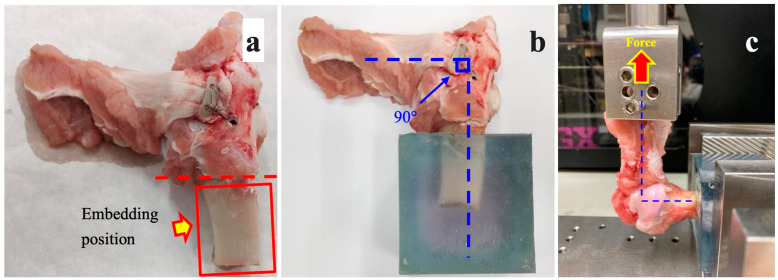
(**a**) The soft tissues on the bone were removed. (**b**,**c**) The infraspinatus muscle specimen was fixed and erected 90 degrees from the humerus during the biomechanical testing.

**Figure 4 bioengineering-10-00565-f004:**
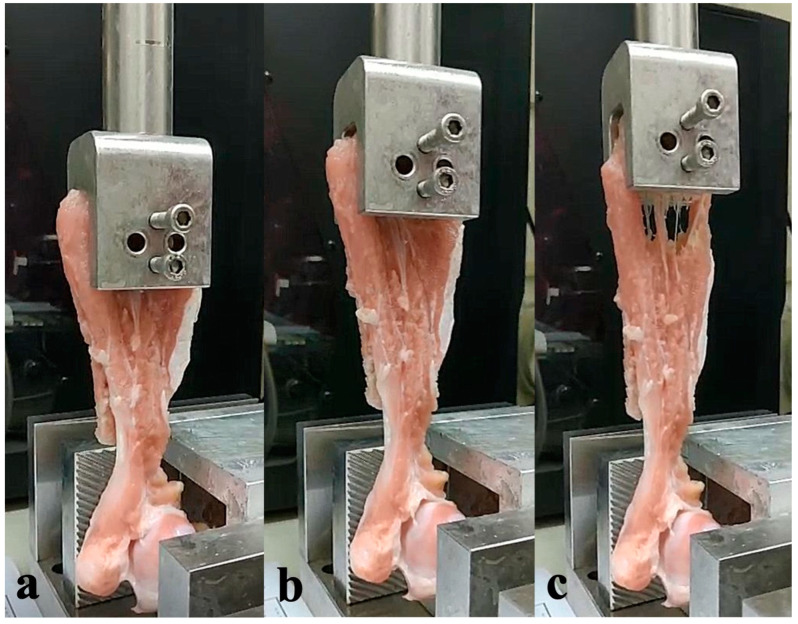
(**a**) Before testing. (**b**) The specimen was pulled at a rate of 75 mm/min. (**c**) Failure was observed at the infraspinatus muscle, but not the enthesis.

**Figure 5 bioengineering-10-00565-f005:**
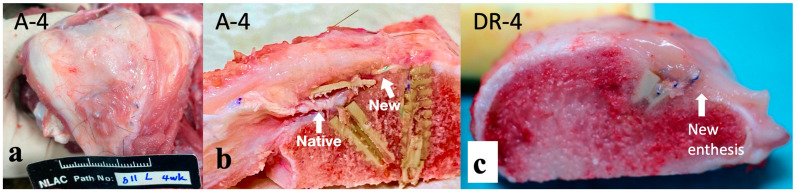
Macroscopic findings of the repaired tendon and bone. (**a**) The specimen before dissection. A well-developed fibrotic appearance was observed in the A-4 group. (**b**) The specimen of the A-4 group after dissection. Retracted native tendon (Native) and well-developed, thicker (7.5 mm) new entheses (New) are observed. (**c**) New enthesis of specimen in DR-4 group has thinner (3.5 mm) appearance than that in A-4 group.

**Figure 6 bioengineering-10-00565-f006:**
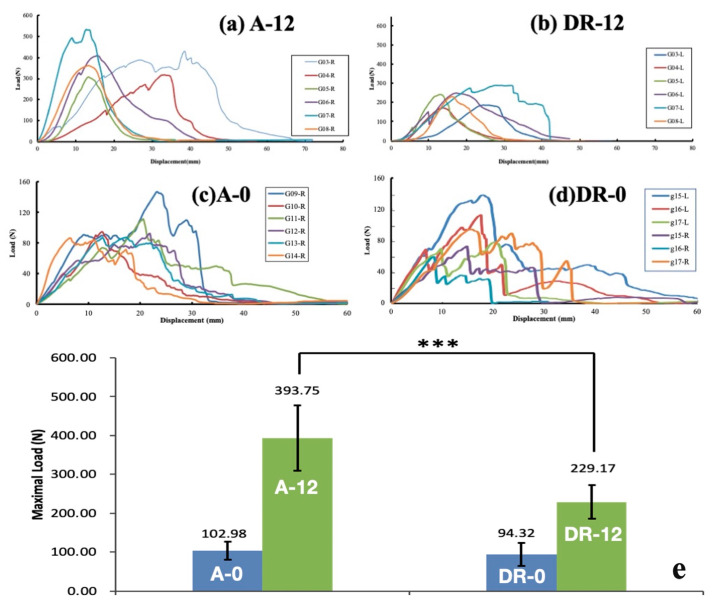
Load-displacement plot of (**a**) A-12 group, (**b**) DR-12 group, (**c**) A-0 group, and (**d**) DR-0 group and (**e**) comparison among the 4 groups. The pull-out strength of A-0 vs. DR-0 and A-12 vs. DR-12 group. The error bar stands for standard deviation (*n* = 6). Statistical significance between A-12 and DR-12 is observed (*** *p* < 0.001, one-way ANOVA for independent samples).

**Figure 7 bioengineering-10-00565-f007:**
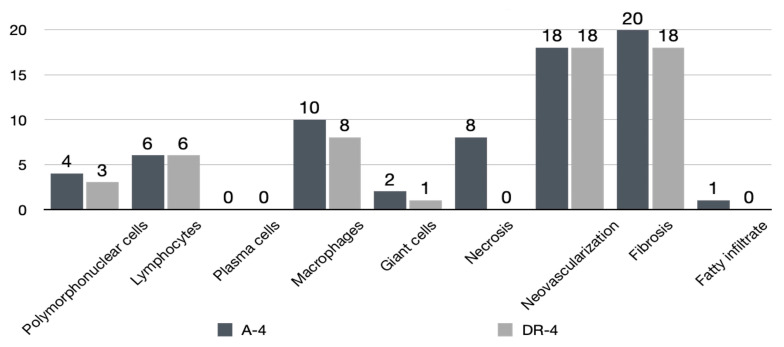
Histopathological cell response and tissue alternation in tendon-to-bone insertion site with (A-4) and without (DR-4) PEEK augment.

**Figure 8 bioengineering-10-00565-f008:**
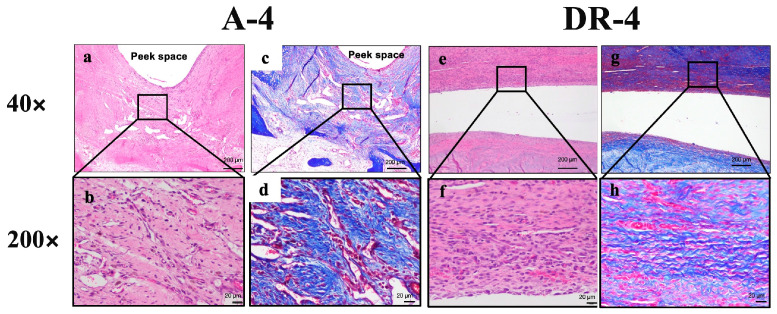
Low- and high-power fields section of both A-4 and DR-4 groups. (**a**,**b**) 40× and 200× of A-4 group under H&E stain and Masson’s trichrome stain (**c**,**d**) showed more sub-acute to sub-chronic inflammatory response, such as fibrin deposition and cell apoptosis, than DR-4 group. (**e**,**f**) H&E stain and (**g**,**h**) Masson’s trichrome stain of DR-4 group.

**Figure 9 bioengineering-10-00565-f009:**
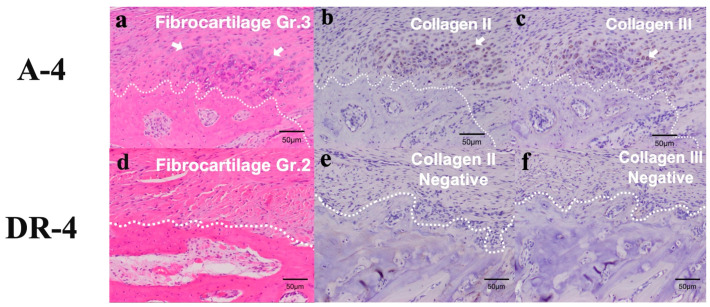
H&E stain and IHC stain results in new footprints area of A-4 group (**a**–**c**) and DR-4 group (**d**–**f**). The dashed border represents the interface between osseous region and soft tissue. Fibrocartilage maturation is obvious in new enthesis area, with positive Collagen II and Collagen III secretion in A-4 group, but not in DR-4 group. (**a**) The fibrocartilage maturation score was Grade 3, where fibrous tissue ingrowth and fibrocartilage cells (arrow) were obvious in new enthesis. (**b**) Positive collagen II signal in fibrocartilage cells (arrow). (**c**) Positive collagen III signal in fibrocartilage cells (arrow). (**d**) The fibrocartilage maturation score was Grade 2 in DR-4 group. (**e**) Negative collagen II signal in fibrocartilage cells. (**f**) Negative collagen III signal in fibrocartilage cells.

**Table 1 bioengineering-10-00565-t001:** Material testing machine characteristics.

Model	Capacity	Speed Accuracy	Load Accuracy	Capacity
HT-2402EC	500 Kgf	±1%	0.01%	0.005~500 mm/min

**Table 2 bioengineering-10-00565-t002:** Scoring system of cell response after surgical repair.

Cell Type/Response	Score				
	0	1	2	3	4
Polymorphonuclear cells	0	1–5/phf ^a^	5–10/phf	Heavy infiltrate	Packed
Lymphocytes	0				
Plasma cells	0				
Macrophages	0				
Giant cells	0	1–2/phf	3–5/phf		Sheets
Necrosis	0	Minimal	Mild	Moderate	Severe

^a^ phf = per high-powered (400×) field.

**Table 3 bioengineering-10-00565-t003:** Scoring system of tissue alternation.

Response	Score				
	0	1	2	3	4
Neovascularization	0	Minimal capillary proliferation, focal, 1–3 buds	Groups of 4–7 capillaries with supporting fibroblastic structures	Broad band of capillaries with supporting structures	Extensive band with supporting fibroblastic structures
Fibrosis	0	Narrow band	Moderately thick band	Thick band	Extensive band
Fatty infiltrate	0	Minimal amount of fat associated with fibrosis	Several layers of fat and fibrosis	Elongated and broad accumulation of fat cells about the implant site	Extensive fat completely surrounding the implant

**Table 4 bioengineering-10-00565-t004:** Scoring system of the overall rating of the surgical impact.

Rating	Score
Minimal or no reaction	(0.0 up to 2.9)
Slight reaction	(3.0 up to 8.9)
Moderate reaction	(9.0 up to 15.0)
Severe reaction	(>15)

**Table 5 bioengineering-10-00565-t005:** Semi-quantitative scoring system of the enthesis maturation.

Grading	C	I	F	T	Definition
1	+				The insertion had continuity without fibrous tissue or bone ingrowth
2	+	+			The insertion had continuity with fibrous tissue ingrowth, but no fibrocartilage cells
3	+	+	+		The insertion had continuity with fibrous tissue ingrowth and fibrocartilage cells, but no tidemark
4	+	+	+	+	The insertion had continuity with fibrous tissue ingrowth, fibrocartilage cells, and a tidemark

C: Continuity, I: Ingrowth, F: Fibrocartilage, T: Tidemark, +: Positive.

**Table 6 bioengineering-10-00565-t006:** Maximal load of separate specimens in the biomechanical test. A: Augmented; DR: Double-row.

Group (*n* = 6)	Average Maximal Load (N)
A-12	393.75 (84.40) *
DR-12	229.17 (43.94)
A-0	102.98 (23.14)
DR-0	94.32 (29.32)

* *p* < 0.001 when compared to DR-12 group.

**Table 7 bioengineering-10-00565-t007:** Cell response and tissue alternation in tendon-to-bone insertion site with and without PEEK augment. A: Augmented; DR: Double-row.

Group	A-4 (*n* = 6)	DR-4 (*n* = 6)
	Score	
Average	99/6 = 16.5	72/6 = 12.0
	Difference between A-4 & DR-4: 16.5 − 12 = 4.5, Slight reaction	

**Table 8 bioengineering-10-00565-t008:** Result of tissue remodeling. A: Augmented; DR: Double-row.

Group	A-4	DR-4	
Sample quantity	6	6	
Enthesis maturation grade	Native footprint part		*p*-value
Grade ≥ 3	3/6 (50.0%)	1/6 (16.7%)	0.545
Median (Min:Max)	2.5 (2:3)	2 (2:3)	0.545
	New footprint part		
Grade ≥ 3	5/6 (83.3%)	3/6 (50.0%)	0.545
Median (Min:Max)	3 (2:3)	2.5 (2:3)	0.242

**Table 9 bioengineering-10-00565-t009:** IHC stain from native footprint part. A: Augmented; DR: Double-row.

Group	A-4	DR-4
Sample quantity	6	6
Native footprint part, bone		
Collagen I (+)	0/6	0/6
Collagen II (+)	0/6	2/6
Collagen III (+)	0/6	0/6
Native footprint part, cartilage		
Collagen I (+)	1/5	1/3
Collagen II (+)	5/5	3/3
Collagen III (+)	5/5	3/3
Native footprint part, tendon		
Collagen I (+)	0/6	0/6
Collagen II (+)	2/6	1/6
Collagen III (+)	2/6	2/6

**Table 10 bioengineering-10-00565-t010:** IHC stain from new footprint part. A: Augmented; DR: Double-row.

Group	A-4	DR-4
Sample quantity	6	6
New footprint part, bone		
Collagen I (+)	0/6	0/6
Collagen II (+)	1/6	3/6
Collagen III (+)	0/6	0/6
New footprint part, cartilage		
Collagen I (+)	1/3	0/1
Collagen II (+)	3/3	1/1
Collagen III (+)	3/3	1/1
New footprint part, tendon		
Collagen I (+)	0/6	0/6
Collagen II (+)	1/6	0/6
Collagen III (+)	4/6	0/6

## Data Availability

Not applicable.

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
