# Peer review of "Global Compressive Loading from an Ultra-Thin PEEK Button Augment Enhances Fibrocartilage Regeneration of Rotator Cuff Enthesis"

_bioengineering, 2023, doi:10.3390/bioengineering10050565_

Round 1
Reviewer 1 Report
This study innovatively designed a button augment combining PEEK plate and suture mechanics to promote rotator cuff repair. This device has certain clinical application value, showing its superiority over the classical double-row technique. Some suggestions and questions are as follows.
1. The advantages and necessity of choosing PEEK as button materials need further be explained in Part Introduction.
2. The study highlights the ultra-thin nature of the PEEK button, which needs to be compared with other studies or applications.
3. The characteristics of PEEK button of global compressive loading are not demonstrated visually in mechanical experiments.
4. The statistical significance should not be simply tested by T-test because of the bivariate factors of technology and time in Figure 6e.
5. Staining should be displayed in the low-power field with the whole section. Staining of all groups is suggested to show for readers better know and compare the differences.
Moderate
Author Response
Reviewer 1
- The advantages and necessity of choosing PEEK as button materials need further be explained in Part Introduction.
Reply: Thanks for the comment. We used PEEK material because it is biologically inert, radiolucent, and resistant to hydrolysis and oxidation.[1-5] We have explained this in the revised manuscript.
- The study highlights the ultra-thin nature of the PEEK button, which needs to be compared with other studies or applications.
Reply: Thanks for the comment. The ultra-thin nature of the PEEK button is a novel design than other bridging materials such as “Regeneten implant”(Smith & Nephew, Andover, MA, USA)”[6] or biceps smash graft[7], which provide only biological augmentation other than mechanical compression. Also, these two applications are newly developed without longer-term follow-up. Hence, we compare the result of the ultra-thin nature of the PEEK button and the commonly used double-row suture technique in this study.
- The characteristics of PEEK button of global compressive loading are not demonstrated visually in mechanical experiments.
Reply: Thanks for the comment. The fine element analysis of the global compressive load of the PEEK button is shown in the pictures below and the revised manuscript (Figure 2d, e). The pressure is dispersed averagely underneath the PEEK button. While the sutures used during rotator cuff repairs are thin, and the stress concentration lies in the suture-tendon area. The PEEK material sustains the shear force of the sutures, converting them into a compression force and providing a homogeneous pressure distribution which could further benefit tendon healing proved in our animal study.
- The statistical significance should not be simply tested by T-test because of the bivariate factors of technology and time in Figure 6e.
Reply: Thanks for the comment. We used one-way ANOVA to determine the difference of load-displacement between 4 groups at different time points (A-0, DR-0, A-12, and DR-12). It is modified in the revised manuscript.
- Staining should be displayed in the low-power fieldwith the whole section. Staining of all groups is suggested to show for readers better know and compare the differences.
Reply: Thanks for the comment. Staining of all groups with low and high-power fields is shown in a new figure (Figure 8) to compare the difference. The previous “Figure 8” is renamed as “Figure 9”.

Reviewer 2 Report
The authors have developed a novel fixation method for rotator cuff repair utilizing an ultra-thin PEEK button used to improve the area and extent of tendon-to-bone compression. The hypothesis is to result in a superior enthesis regeneration characterized by greater fibrocartilage formation and improved collagen fiber organization in an acute rotator cuff tear goat model.
Interesting, well-constructed study. The experimental design is presented in a comprehensible way. Biomechanical testing completes the work. It is good that the histological workup was so cleanly processed. The conclusion is critical and comprehensible.
They show that their results provides superior load-displacement property to conventional DR technique. Though not significantly different, they see a trend toward better fibrocartilage maturation and more collagen III secretion in the PEEK augmentation group than the conventional DR repair group.
s.above
Author Response
Reply: Thanks for the comment.

Reviewer 3 Report
Overall, the study appears to be well-designed and comprehensive in its evaluation of the PEEK button as a novel device to improve tendon-to-bone compression area. The division of goats into different groups based on the duration of the detachment and use of PEEK augment or double-row technique adds clarity to the study and allows for better comparisons of the results. Overall, I found the manuscript to be well-organized and informative. However, there are several major revisions that should be made before the manuscript can be accepted for publication.
1. In terms of the limitations of the study, the small sample size of goats may limit the generalizability of the findings to humans. Additionally, the short-term evaluation of the 4-week and 0-week groups may not fully capture the long-term effects of the PEEK button.
2. Introduction need to be expanded to elaborate more about the PEEK and its contribution toward analysis.
3. Provide more information about the rationale behind the study design, such as why three different time points were chosen and why two different techniques were used.
4. Consider providing more details about the surgical procedures, such as the size of the anchors used and the specific knots employed.
5. Clarify the purpose of the PEEK augment, and provide more information about why it was used in some groups but not others.
6. Consider adding a sentence or two about the expected outcomes of the study based on the surgical techniques and groups being compared.
7. The description of the IHC assessment is well-written and provides a clear understanding of the steps involved in sample preservation and processing. However, the section seems to describe histological evaluation rather than immunohistochemistry. If the authors did perform IHC analysis, it should be mentioned in this section, along with the type of antibody used and the staining protocol.
8. The results should be described in more detail, including the specific outcomes of the A-12 and DR-12 groups, and how they compare to the A-4 and DR-4 groups.
9. The interpretation of the results should be clarified, with an emphasis on how the PEEK augment contributes to enthesis healing and how it compares to other repair techniques.
10. The paragraph on the formation of new enthesis and the use of biologic materials could be more clearly written and structured.
11. The paragraph on the mechanism of PEEK augment healing potential could be more concise and focused.
12. The paragraph on the failure modes of A-0 and DR-0 groups requires more clarity.
13. There is lack of literature thus author is suggested to add some relevant literature such as:
Journal of Bio-and Tribo-Corrosion, 7, 1-48 https://doi.org/10.1007/s40735-021-00501-y, Coatings, 12(10), 1459 https://doi.org/10.3390/coatings12101459
Minor checking and revision required
Author Response
Reviewer 3
- In terms of the limitations of the study, the small sample size of goats may limit the generalizability of the findings to humans. Additionally, the short-term evaluation of the 4-week and 0-week groups may not fully capture the long-term effects of the PEEK button.
Reply: Thanks for the comment. Indeed, the limitation of the study is the short-term evaluation of this novel design. According to Gerber et al.[8], they also used 18 alpine sheep (6 sheep per group) to test the effects of anabolic steroids on muscle atrophy. Muscle biopsy specimens were taken at 16 and 22 weeks after treatment. Wang et al. also used 12 alpine sheep to see the muscle edema of retraction after greater tuberosity osteotomy at 2 and 4 weeks postoperatively.[9] Therefore, we used 18 male black goats in our series and performed the histological and biomechanical tests at 4 weeks and 12 weeks postoperatively.
- Introduction need to be expanded to elaborate more about the PEEK and its contribution toward analysis.
Reply: Thanks for the comment. We used PEEK material because it is biologically inert, radiolucent, and resistant to hydrolysis and oxidation.[1-5] We have explained in the revised manuscript.
- Provide more information about the rationale behind the study design, such as why three different time points were chosen and why two different techniques were used.
Reply: Thanks for the comment. We choose 0-week to see the initial fixation strength of the PEEK augment and compared them with the commonly used double-row technique in clinical practice. At 4 weeks, we tested the cell responses and tissue alternations as they were during the inflammatory phase after the surgical treatment. At 12 weeks, we tested the biomechanical characteristics of both groups (Figure 6) to see the fixation strength after tendon healing.
- Consider providing more details about the surgical procedures, such as the size of the anchors used and the specific knots employed.
Reply: Thanks for the comment. We used one 5.5mm Healix Peek anchor ((DePuy Mitek, Raynham, MA) for the medial row and one 4.75mm PEEK SwiveLock suture anchor (Arthrex, Naples, FL) for the lateral row fixation. Horizontal mattress sutures were tied using five alternating half-hitch knots to reproduce arthroscopic knot configurations. This is added in the revised manuscript.
- Clarify the purpose of the PEEK augment and provide more information about why it was used in some groups but not others.
Reply: Thanks for the comment. The object of this study is to demonstrate the safety and the compressive effective of the PEEK augment and how this new design can help to improve enthesis regeneration. Therefore, we have compared the PEEK with DR technique, which is commonly used during clinical practice. And we proved that it is safe and would result in a superior enthesis regeneration characterized by greater fibrocartilage formation and improved collagen fiber organization in an acute rotator cuff tear goat model. This PEEK button augmentation would lead to higher biomechanical stiffness when compared with a simple repair with sutures.
- Consider adding a sentence or two about the expected outcomes of the study based on the surgical techniques and groups being compared.
Reply: Thanks for the comment. We add the following sentence in the revised manuscript “It represented the commonly used double-row suture-bridge technique popularized in clinical practice and provided the expected fixation strength.”
- The description of the IHC assessment is well-written and provides a clear understanding of the steps involved in sample preservation and processing. However, the section seems to describe histological evaluation rather than immunohistochemistry. If the authors did perform IHC analysis, it should be mentioned in this section, along with the type of antibody used and the staining protocol.
Reply: Thanks for the comment. Both histological evaluation and the immunohistochemistry stain are presented in the new “Figure 8” in the revised manuscript. The staining protocol was added in the reference in the revised manuscript.[10]
- The results should be described in more detail, including the specific outcomes of the A-12 and DR-12 groups, and how they compare to the A-4 and DR-4 groups.
Reply: Thanks for the comment. This result implied that the PEEK augment significantly improved tendon healing quality and provided superior load displacement in the A-12 group than the DR-12 group at a 12-week time interval (393.75 (84.40) N in the A-12 group and 229.17 (43.94) N in the DR-12 group, P<0.001)), which was not observed at time zero (A-0 and DR-0 group, maximum load, 102.98 (23.14) N, and 94.32 (29.32) N, P =0.291). This is added in the revised manuscript. However, we did not perform biomechanical testing at A-4 and DR-4 groups at 4 weeks post-operation because it normally takes at least 3 months until functional recovery after rotator cuff repair.[11]
- The interpretation of the results should be clarified, with an emphasis on how the PEEK augment contributes to enthesis healing and how it compares to other repair techniques.
Reply: Thanks for the comment. According to our data, the PEEK augment provided homogeneous pressure distribution on the cuff footprint, contributing to better fibrocartilage growth with more collagen III secretions in the new footprint of the A-4 group than in the DR-4 group. At 12 weeks, the PEEK augment group (A-12) had improved tendon healing quality and superior load-displacement than the DR-12 group. The above sentence is added in the revised manuscript.
- The paragraph on the formation of new enthesis and the use of biologic materials could be more clearly written and structured.
Reply: Thanks for the comment. The sentence is rewritten in the revised manuscript.
- The paragraph on the mechanism of PEEK augment healing potential could be more concise and focused.
Reply: Thanks for the comment. The sentence is rewritten in the revised manuscript.
- The paragraph on the failure modes of A-0 and DR-0 groups requires more clarity.
Reply: Thanks for the comment. The sentence is rewritten in the revised manuscript.
- There is lack of literature thus author is suggested to add some relevant literature such as: Journal of Bio-and Tribo-Corrosion, 7, 1-48 https://doi.org/10.1007/s40735-021-00501-y, Coatings, 12(10), 1459 https://doi.org/10.3390/coatings12101459
Reply: Thanks for the comment. The reference is added in the revised manuscript.
References
- Di Benedetto, P.; Gorasso, G.; Beltrame, A.; Mancuso, F.; Buttironi, M.M.; Causero, A. Clinical and radiological outcomes with PEEK suture anchors used in rotator cuff repair: our experience confirm that a perianchor fluid signal on RM does not affect clinical outcome at one year of follow up. Acta Bio Medica: Atenei Parmensis 2020, 91.
- Dhawan, A.; Ghodadra, N.; Karas, V.; Salata, M.J.; Cole, B.J. Complications of bioabsorbable suture anchors in the shoulder. Am J Sports Med 2012, 40, 1424-1430.
- Güleçyüz, M.F.; Mazur, A.; Schröder, C.; Braun, C.; Ficklscherer, A.; Roßbach, B.P.; Müller, P.E.; Pietschmann, M.F. Influence of Temperature on the Biomechanical Stability of Titanium, PEEK, Poly-L-Lactic Acid, and β–Tricalcium Phosphate Poly-L-Lactic Acid Suture Anchors Tested on Human Humeri In Vitro in a Wet Environment. Arthroscopy: The Journal of Arthroscopic & Related Surgery 2015, 31, 1134-1141.
- Christensen, J.; Fischer, B.; Nute, M.; Rizza, R. Fixation strength of polyetheretherketone sheath-and-bullet device for soft tissue repair in the foot and ankle. The Journal of Foot and Ankle Surgery 2018, 57, 60-64.
- Kurtz, S.M.; Devine, J.N. PEEK biomaterials in trauma, orthopedic, and spinal implants. Biomaterials 2007, 28, 4845-4869.
- Thon, S.G.; Belk, J.W.; Bravman, J.T.; McCarty, E.C.; Savoie, F. Regeneten bio-inductive collagen scaffold for rotator cuff tears: indications, technique, clinical outcomes, and review of current literature. Ann Joint 2020, 5, 41.
- Endell, D.; Rüttershoff, K.; Scheibel, M. Biceps Smash Technique: Biceps Tendon Autograft Augmentation for Arthroscopic Rotator Cuff Reconstruction. Arthroscopy Techniques 2023, 12, e383-e386.
- Gerber, C.; Meyer, D.C.; Fluck, M.; Benn, M.C.; von Rechenberg, B.; Wieser, K. Anabolic Steroids Reduce Muscle Degeneration Associated With Rotator Cuff Tendon Release in Sheep. Am J Sports Med 2015, 43, 2393-2400, doi:10.1177/0363546515596411.
- Wang, S.; Lädermann, A.; Chiu, J.; Nabergoj, M.; Ho, S.W.; Brigitte, v.R.; Bothorel, H.; Lädermann, L.; Kolo, F. Muscle Edema of Retraction and Pseudo–Fatty Infiltration After Traumatic Rotator Cuff Tears: An Experimental Model in Sheep. Orthopaedic Journal of Sports Medicine 2023, 11, 23259671231154275.
- Van De Vlekkert, D.; Machado, E.; d’Azzo, A. Analysis of generalized fibrosis in mouse tissue sections with Masson’s trichrome staining. Bio-protocol 2020, 10, e3629-e3629.
- Charousset, C.; Grimberg, J.; Duranthon, L.D.; Bellaïche, L.; Petrover, D.; Kalra, K. The time for functional recovery after arthroscopic rotator cuff repair: correlation with tendon healing controlled by computed tomography arthrography. Arthroscopy: The Journal of Arthroscopic & Related Surgery 2008, 24, 25-33.

Round 2
Reviewer 1 Report
The author well addressed the questions except:
1. It appears the sections selected in each group are different from each other in Fig 8. Please explain this. And the scale bar is missing here.
2. Please provide the H&E stain and IHC stain results in the DR-4 group in contrast to the A-4 group in Fig 9.
The language should be improved. For example, there should be space between the number and the unit, e.g., 50 μm instead of 50μm.
Author Response
It appears the sections selected in each group are different from each other in Fig 8. Please explain this. And the scale bar is missing here.
Reply: Thanks for the comment. “Figure 8” is revised with the same group of slices, and the scale bar is added in the revised manuscript.
- Please provide the H&E stain and IHC stain results in the DR-4 group in contrast to the A-4 group in Fig 9
Reply: Thanks for the comment. The H&E stain and IHC stain in the DR-4 group is provided in the revised manuscript (Figure 9d, e, f).
Comments on the Quality of English Language
The language should be improved. For example, there should be space between the number and the unit, e.g., 50 μm instead of 50μm.
Reply: Thanks for the comment. The space between the number and the unit is corrected in the revised manuscript.

Reviewer 3 Report
The author has addressed all the queries and has improved the manuscript. It can be accepted now.
Author Response
Thanks for the comment.